# Unfolding the mitochondrial genome structure of green semilooper (*Chrysodeixis acuta* Walker): An emerging pest of onion (*Allium cepa* L.)

**Soumia P. S.** [1]*, **Dhananjay V. Shirsat**[1], **Ram Krishna**[1], **Guru Pirasanna Pandi G.**[2], **Jaipal S. Choudhary**[3], **Naiyar Naaz**[3], **Karuppaiah V.**[1], **Pranjali A. Gedam**[1], **Anandhan S.**[1]*, **Major Singh**[1]

**1** ICAR-Directorate of Onion and Garlic Research (DOGR), Pune, Maharashtra, India, **2** ICAR-National Rice Research Institute (NRRI), Cuttack, Odisha, India, **3** ICAR- Research Complex for Eastern Region (RCER), Research Centre, Ranchi, India

* soumiaps@gmail.com (SPS); anandhans@gmail.com (AS)

**Data Availability Statement:** The sequencing data underlying this study have been submitted to NCBI gene bank with accession number OL892047.

## Abstract

Onion is the most important crop challenged by a diverse group of insect pests in the agricultural ecosystem. The green semilooper (*Chrysodeixis acuta* Walker), a widespread tomato and soybean pest, has lately been described as an emergent onion crop pest in India. *C. acuta* whole mitochondrial genome was sequenced in this work. The circular genome of *C. acuta* measured 15,743 base pairs (bp) in length. Thirteen protein-coding genes (PCGs), 22 *tRNA* genes, two *rRNA* genes, and one control region were found in the 37 sequence elements. With an average 395 bp gene length, the maximum and minimum gene length observed was 1749 bp and 63 bp of *nad5* and *trnR*, respectively. Nine of the thirteen PCGs have (ATN) as a stop codon, while the other four have a single (T) as a stop codon. Except for *trnS1*, all of the *tRNAs* were capable of producing a conventional clover leaf structure. Conserved *ATAGA motif* sequences and *poly-T* stretch were identified at the start of the control region. Six overlapping areas and 18 intergenic spacer regions were found, with sizes ranged from 1 to 20 bp and 1 to 111 bp correspondingly. Phylogenetically, *C. acuta* belongs to the *Plusiinae* subfamily of the *Noctuidae* superfamily, and is closely linked to *Trichoplusia ni* species from the same subfamily. In the present study, the emerging onion pest *C. acuta* has its complete mitochondrial genome sequenced for the first time.

## Introduction

Onion (*Allium cepa* L.) of the family *Alliaceae* has been cultivated for more than 5000 years [1]. Though onion is one of the important vegetables, it is also being used as a spice for flavoring various cuisines globally [2–4]. Being a bulbous crop, onion is mainly grown for its bulbs. However, the green leaves are also consumed worldwide as raw, cooked, semi-cooked, or in processed forms as a condiment, nutritional, nutraceutical, and medicinal purposes [3, 5, 6].

**Funding:** The present study is a part of in-house project, Reference code "IXX14062." The financial support is given by ICAR- Directorate of Onion and Garlic Research, Pune. The funders had no role in study design, data collection and analysis, decision to publish, or preparation of the manuscript.

**Competing interests:** All the authors declare that there is no competing conflict of interest.

India is the second-largest producer (26738.000 MT), consumer, and exporter of onion globally (https://www.fao.org/faostat/en/#data/QCL). But unfortunately, the productivity per hectare is very low compared to countries where onion is grown commercially. The onion agro-ecosystem has a highly diverse insect pest complex, among which onion thrips, *Thrips tabaci* is the key pest of onion [7]. Besides thrips, several defoliators of the order lepidoptera are found at damaging levels in onions causing monetary loss globally [1, 2]. These insect pests affect almost every stage of the onion and cause a potential yield loss of 20 to 90% in onion production [8, 9]. Insects can coevolve with the host plants, a relatively frequent phenomenon attributed to the prevailing environmental condition and host plant availability [10–12]. Subsequently, insects pose a serious threat by expanding their geographic distribution and host range.

Green semilooper *Chrysodeixis acuta* Walker, an important established pest of soybean and tomato across India, is also becoming an emerging onion pest recently. Onion is commonly intercropped with soybean in India's southern and central agricultural zones to limit insect pests by colonizing biocontrol agents [13]. *C. acuta* was recorded in onions for the first time in Maharashtra, India, in 2017–2018 [14]. The glossy green larvae are highly polyphagous, multivoltine species with huge potential to spread to new locations and adapt to new climatic or ecological conditions, which is likely why it has recently infested onions [14]. *C. acuta* has a wide range of hosts, including various vegetables, grains, and fruit crops [15–17]. *C. acuta*, like most defoliators, feeds on onion leaves; early instars scrape the leaves' mesophyll tissue, leaving papery white structures, while later instars bore enormous feeding holes. Though this pest currently causes minimal damage to onion crops, significant defoliation by these caterpillars is expected where the soybean-onion cropping system is practiced [14]. Excessive defoliation can lower the net photosynthetic area, resulting in reduction of bulb size.

There is very little information available on the geographic variability and genetic structure of *C. acuta*. *Chrysodeixis* belongs to the *Plusiinae* subfamily and the *Noctuidae* family of moths. This subfamily is taxonomically compact and moderately large. More than 500 species have been identified globally [18] with 59 species belonging to 25 genera and three tribes [19] Nonetheless, the NCBI GenBank (https://www.ncbi.nlm.nih.gov/) only contains mitochondrial genome data only for four species from plusiinae subfamily: *Diachrysia nadeja* (MT916722), *Trichoplusia ni* (MK714850), *Ctenoplusia albostriata* (MN495624), and *Ctenoplusia limbirena* (KM244665). However, no pest species from the genus *Chrysodeixis* was sequenced to date even though some of the economically important pests under this genus are *Chrysodeixis chalcites*, *Chrysodeixis eriosoma*, *Chrysodeixis includens*, and *Chrysodeixis acuta*.

Plusiinae subfamily larvae are usually referred to as semiloopers. They are distinguished by 3 pairs of abdominal legs (prolegs) with biordinal crochets. In contrast, adult moths are robust, small to medium-sized, with a characteristic metallic spot in the center of the forewing. Morphological techniques cannot reliably distinguish these pests. However, taxonomic identification of the particular pest species is the fundamental step to device a suitable management strategy [20]. As a result, the mitochondrial (*mt*) genome of insects is frequently utilized as a molecular marker to disclose basic information at the genomic level for phylogenetic inference, molecular evolution, identification of species, geographic distribution, and population dynamics [21, 22]. Therefore, in the present study, the whole *mt* genome of *C. acuta* was assembled for the first time using next-generation sequencing (NGS), which might provide a solution for species evolution and phylogeny. Furthermore, the extensive genetic information obtained from this investigation regarding this new onion pest may aid in devising effective control or preventative strategies.

## Materials and methods

### Sample collection

Green semilooper, *Chrysodeixis acuta* larvae were collected from onion plants at the Indian Council of Agriculture Research—Directorate of Onion and Garlic Research (ICAR-DOGR), situated at the (latitude: 27˚19'00.2 N, longitude: 82˚25'00.1 E, 553.8 meters above sea level) Pune, Maharashtra, India. The insects were initially reared in the laboratory. After completion of the life cycle, newly emerged adults from a single egg mass were collected, one pair was preserved in the insect repository of the Center with voucher specimen number: DOGR Voucher 15 following standard procedures. To further confirm the species identity, larval specimens were analyzed by 'DNA barcoding' at ICAR-DOGR, Pune, India and the sequence was submitted under the accession number MT644267 in NCBI GenBank and GBMNC55083-20 in Barcode of Life Data System (BOLD).

### Sample preparation and DNA extraction

Laboratory reared fully grown fourth instar larvae of *C. acuta* were kept in a refrigerator at -20˚C in 100 percent ethanol until the experiment. According to the manufacturer's instructions, total genomic DNA was extracted using the DNeasy Blood and Tissue Kit (QIAGEN, Germany). The quality and amount of extracted DNA were assessed using a 1% agarose gel and with SmartSpec 3000 UV/Visible Spectrophotometer at 260 and 280 nm (Bio-Rad, Hercules, California, USA). By referring to the REPLI-g Mitochondrial DNA Kit (QIAGEN, Germany) protocol, mitochondrial DNA was isolated from genomic DNA.

### Sequencing and mitogenome analysis

The *mt* genome was sequenced using the Illumina Next Seq 500 sequencing platform and Trimmomatic (v0.38) [23] was used to eliminate sequences of adaptor, ambiguous reads (reads having >5% unidentified nucleotides), and junk sequences (reads having > 10% quality threshold (QV) 20 phred score) from sequenced raw data to get high-quality clean reads. The high-quality reads aligned to the reference sequences with the help of BWA MEM (version 0.7.17) [24]. The consensus sequence was extracted using SAM toolsmpileup [25]. The consensus sequence was employed to identify protein-coding and RNA genes in the sample. The MITOS algorithm was used for genes prediction from the invertebrates' mitochondrial genome [26].

The protein-coding and *rRNA* genes were manually annotated and verified comparing them with four mitogenome sequences of *Diachrysia natija* (MT916722), *Trichoplusia ni* (MK714850), *Ctenoplusia albostriata* (MN495624), and *Ctenoplusia limbirena* (KM244665) of subfamily *Plusiinae* available in GenBank. Using tRNAscanSEv 2.0 and a 15.0 covariance limit value, the Mito/Chloroplast model was utilized to recheck the MITOS downloaded two-dimensional *tRNA* structures [27]. The CGview server was used to generate the circular map of the complete *mt* genome, GC concentration, and GC skew [28]. Finally, protein-coding sequence regions of mitochondria were assembled and aligned in Mega version X using ClustalW with basic parameters [29].

The A+T contents, codon use, and relative synonymous codon usage (RSCU) of the PCGs were analyzed by using MEGA version X. The GC and AT skews were calculated utilizing the formulas (GC skew = [G-C] / [G+C] and AT skew = [A-T] / [A+T]) given by Perna and Kocher [30]. The overlapping and intergenic spacer region between genes were manually calculated. The synonymous (Ks) and nonsynonymous substitution rates (Ka) for each Protein-coding gene were calculated using the DnaSP 6.0 and Jukes-Cantor adjusted Ka/Ks (JKa/

JKs) software programmes [31]. The control region tandem repeats were predicted by using Tandem Repetitions Finder tool with default settings [32]. The assembled and annotated *C. acuta* mitogenome has been submitted under the accession number OL892047 in NCBI GenBank.

## Phylogenetic analysis

All of the available complete *Nocutidae* mitogenomes (55mitogenomes) were chosen for the phylogenetic study. One of the 55 *Nocutidae* mitogenomes was sequenced in this study, and the rest samples were obtained from the NCBI database.

The phylogenetic study used concatenated nucleotide sequences from 13 PCGs datasets. The PCG was aligned using MAFFT 7 (https://mat.cbrc.jp/alignment/server/) based on codons for amino acids [33]. Similarly, to remove ambiguously aligned sites from PCG alignments, GBlocks v.0.91b (http://molevol.cmima.csic.es/castresana/Gblocks/Gblocksdocumentation.html) was used [34]. MEGA 10.0 was used as the final quality check for all of the alignments. All gene alignments were concatenated using PhyloSuite 1.2.1 [35]. PCG123 matrix with single sequences in the following order: *nad2*, *cox1*, *cox2*, *atp8*, *atp6*, *cox3*, *nad3*, *nad5*, *nad4*, *nad4L*, *nad6*, *cytb*, and *nad1*. For phylogenetic reconstruction, these thirteen PCGs were concatenated. The best partitioning schemes were chosen using Partition Finder 2.1.1 with the Bayesian Information Criterion (BIC) and greedy algorithm (www.phylo.org) [36]. According to BIC, the GTR+I+G model was ideal for nucleotide alignment analysis. PHYML online web server utilized the optimum substitution model obtained from Model Test to infer maximum likelihood (ML) analysis, and model parameter values were calculated [37]. Bayesian analyses were performed by utilizing Markov Chain Monte Carlo (MCMC) method with software, Mr. Bayes v.3.1.2 [38] having two independent runs of $2 \times 10^6$ generations with four chains with trees sampled each $1000^{th}$ generation. Mr. Bayes' "sump" command was used to evaluate the similar values for every post-analysis trees and the convergence and burn-in parameters. The top 100 trees from each run were utilized for burn-in, and the remaining trees were used to form a consensus tree with a 50% majority rule.

## Results and discussion

### Organization of the genome

The semilooper's circular genome was 15,399 nucleotides in length. There are 37 sequence elements in total, comprising 13 protein-coding genes (PCGs) which include *Cytochrome c oxidase*, *NADH dehydrogenase*, *Cytochrome B*, *ATPase*, and two *rRNA* genes with 22 *tRNA* genes, and a control region. Every gene having an average gene length of 395 bp; the maximum and minimum gene lengths were 1749 and 63 bp, respectively (Fig 1). The mitochondrial genome of *C. acuta* was sequenced, and its molecular and phylogenetic features were studied in this work. In general, the mitogenome of insects are circular double -stranded molecule of 14–19 kb size which includes 37 genes: 13 PCGs, 2 *rRNA* genes and 22 *tRNA* genes [39, 40]. Gene content, order, and orientation are identical to those of reported noctuid mitogenomes and are typical of Lepidoptera [41, 42]. The semilooper's circular *mt* genome measured 15,399 bp in size, slightly bigger than *Trichoplusia ni* (15,239 bp), which belongs to the same *Plusiinae* subfamily [43]. Similarly, the *mt* genome of green semilooper is closely related to the other defoliator pests infesting onions, such as *S. exigua* (15,365 bp) and *S. litura* (15,374 bp) [44, 45]. *C. acuta* is the third insect pest from the *Plusiinae* subfamily to have its whole *mt* genome sequenced after *T. ni* [43] and *Macdunnoughia hybrida* [42].

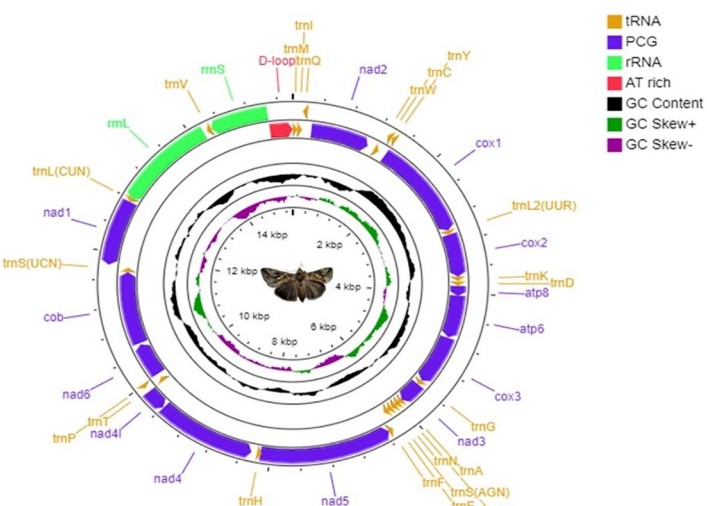

**Fig 1. Mitochondrial genome map of *Chrysodeixis acuta*.** From outer to inner, the 1st circle shows the gene map (PCGs, rRNA, tRNAs & CR) and tRNA genes are abbreviated by one letter symbols according to the IUPAC-IUB single-letter amino acid codes. The 2nd circle shows the GC content and the 3rd shows GC skew calculated as (G-C)/(G+C). GC content and GC skew are plotted as the deviation from the average value of the entire sequence.

## Protein-coding genes (PCGs)

The *mt* genome of *C. acuta* has 13 genes coding protein scattered in a circular chromosome. These PCGs contain three *cytochrome c oxidase* (*cox1*, *cox2*, and *cox3* genes), seven *NADH dehydrogenase* (*nad1*, *nad2*, *nad3*, *nad4*, *nad5*, *nad6*, *and nad4l* genes), one cytochrome B (*cytB* gene), and two *ATPase* (*atp6* and *atp8* genes) (Table 1). The H-strand included nine of the 13 PCGs (*cox1*, *cox2*, *cox3*, *nad2*, *nad3*, *nad6*, *atp6*, *atp8*, and *cytB*), whereas the L-strand had the remaining four (*nad1*, *nad4*, *nad4l*, *and nad5*). In *C. acuta*, four start codons (ATA, ATT, ATG, and ATC) were found. Seven genes (*cox2*, *cox3*, *atp6*, *nad1*, *nad4*, *nad4l*, and *cytB*) adopted ATG as the start codons, 3 genes (*cox1*, *atp8* and *nad5*) used ATT, two (*nad2* and *nad6*) used ATA, while ATC by single gene (*nad3*). Nine PCGs employed the TAA termination codon (*cox3*, *atp6*, *atp8*, *nad1*, *nad3*, *nad4l*, *nad5*, *nad6*, and cytB). The incomplete termination codon T was found in four genes: *cox1*, *cox2*, *nad2*, and *nad4* (Table 1). Table 2 summarizes the values for the relative synonymous codon use (RSCU) and amino acid usage in the PCGs of *C. acuta*. Phenylalanine (Phe, F), Leucine (Leu, L), Isoleucine (Ile, I), Asparagine (Asn, N), and Serine (Ser, S) were found to be the five most commonly occurring amino acids, whereas Cysteine (Cys, C), and Arginine (Arg, R) were found to be the rarest (Table 2). The protein-coding genes are found scattered on both heavy and light strands, covering 71.93% of the total *mt* genome of *C. acuta*, and—11,077 bp long. These PCGs include cytochrome c oxidase, NADH dehydrogenase, Cytochrome B, and ATPase, commonly found in most insect species belonging to Lepidopteran order [21, 46–48].

Insect species like *Ctenoplusia albostriata* [49], *Diachrysia nadeja* [50], *Trichoplusia ni* [43], *Laelia suffusa* [51], *Cerura menciana* [22] and *Eudocima salaminia* [48] all have the same genes with little differences in size and location. The start codons for the *cox1* gene in lepidopteran insects are not uniform [52], which has been widely debated and has long been a source of contention [53]. The canonical start codon ATN—was used to initiate all of *C. acuta's* protein-coding genes. The initial codon for *cox1* was ATT, instead of "CGA" utilized in several insect species [54]. In *C. acuta*, two termination codons were discovered. Four genes *cox1*, *cox2*, *nad2*, and *nad4*, used the incomplete termination codon "T"; nevertheless, other genes

**Table 1. Summary table for characteristics of the complete mitogenome of *Chrysodeixis acuta*.**

| Name | Start position | Stop position | Strand | Length | Intergenic Spacers | Anticodon | Start codon | stop codon |
|---|---|---|---|---|---|---|---|---|
| trnM | 1 | 68 | + | 68 | | atg | | |
| trnI | 69 | 135 | + | 67 | 0 | atc | | |
| trnQ | 133 | 201 | - | 69 | -2 | caa | | |
| nad2 | 276 | 1149 | + | 874 | 74 | | ata | t |
| trnW | 1261 | 1327 | + | 67 | 111 | tga | | |
| trnC | 1320 | 1384 | - | 65 | -7 | tgc | | |
| trnY | 1386 | 1450 | - | 65 | 1 | tac | | |
| cox1 | 1443 | 2988 | + | 1546 | -7 | | att | t |
| trnL2 | 2989 | 3055 | + | 67 | 0 | tta | | |
| cox2 | 3056 | 3737 | + | 682 | 0 | | atg | t |
| trnK | 3738 | 3808 | + | 71 | 0 | aag | | |
| trnD | 3818 | 3886 | + | 69 | 9 | gac | | |
| atp8 | 3887 | 4048 | + | 162 | 0 | | att | taa |
| atp6 | 4042 | 4719 | + | 678 | -6 | | atg | taa |
| cox3 | 4730 | 5515 | + | 786 | 10 | | atg | taa |
| trnG | 5521 | 5585 | + | 65 | 5 | gga | | |
| nad3 | 5595 | 5939 | + | 345 | 9 | | atc | taa |
| trnA | 5939 | 6003 | + | 65 | -1 | gca | | |
| trnR | 6004 | 6066 | + | 63 | 0 | cga | | |
| trnN | 6067 | 6133 | + | 67 | 0 | aac | | |
| trnS1 | 6137 | 6202 | + | 66 | 3 | agc | | |
| trnE | 6203 | 6268 | + | 66 | 0 | gaa | | |
| trnF | 6298 | 6364 | - | 67 | 29 | ttc | | |
| nad5 | 6365 | 8113 | - | 1749 | 0 | | att | taa |
| trnH | 8114 | 8179 | - | 66 | 0 | cac | | |
| nad4 | 8252 | 9590 | - | 1339 | 72 | | atg | t |
| nad4l | 9597 | 9887 | - | 291 | 6 | | atg | taa |
| trnT | 9890 | 9955 | + | 66 | 2 | aca | | |
| trnP | 9956 | 10020 | - | 65 | 0 | cca | | |
| nad6 | 10028 | 10561 | + | 534 | 7 | | ata | taa |
| cytB | 10581 | 11732 | + | 1152 | 19 | | atg | taa |
| trnS2 | 11735 | 11800 | + | 66 | 2 | tca | | |
| nad1 | 11820 | 12758 | - | 939 | 19 | | atg | taa |
| trnL1 | 12760 | 12828 | - | 69 | 1 | cta | | |
| rrnL | 12808 | 14166 | - | 1359 | -20 | | | |
| trnV | 14215 | 14279 | - | 65 | 48 | gta | | |
| rrnS | 14280 | 15060 | - | 781 | 0 | | | |
| Control region | 15061 | 15399 | + | 339 | 0 | | | |

were terminated by TAA. Our findings are consistent with those of Liu *et al.* [55], Dai *et al.* [22], Dai *et al.* [56], Li *et al.* [57] Chen *et al.* [58] and Riyaz *et al.* [48] who also reported the presence of single "T" as a termination codon for the *cox1* and *cox2* genes in the majority of Lepidopteran species. This incomplete termination codon "T" in lepidopteran *mt* genes might get polyadenylated to TAA codon during translation [46, 47, 59]. Numerous studies have demonstrated that the nucleotides 'A' and 'T' are generally overrepresented in metazoan mitogenomes, which causes bias in the corresponding encoded amino acids [21]. Codons having 'A' or 'T' at the third prime position were found to be over used in this study when compared to

**Table 2. Codon usage and relative synonymous codon usage (RSCU) within *Chrysodeixis acuta* mitochondrial genome.**

| Codon | Count | RSCU | Codon | Count | RSCU | Codon | Count | RSCU | Codon | Count | RSCU |
|---|---|---|---|---|---|---|---|---|---|---|---|
| UUU(F) | 26.3 | 1.92 | UCU(S) | 10.2 | 3.3 | UAU(Y) | 14.2 | 1.88 | UGU(C) | 2.3 | 2 |
| UUC(F) | 1.2 | 0.08 | UCC(S) | 0.6 | 0.2 | UAC(Y) | 0.9 | 0.12 | UGC(C) | 0 | 0 |
| UUA(L) | 35.4 | 5.27 | UCA(S) | 4.9 | 1.59 | UAA(*) | 0.7 | 2 | UGA(W) | 7.1 | 1.96 |
| UUG(L) | 0.6 | 0.09 | UCG(S) | 0.1 | 0.02 | UAG(*) | 0 | 0 | UGG(W) | 0.2 | 0.04 |
| CUU(L) | 2.9 | 0.44 | CCU(P) | 6.5 | 2.69 | CAU(H) | 4.8 | 1.85 | CGU(R) | 1.4 | 1.33 |
| CUC(L) | 0 | 0 | CCC(P) | 0.8 | 0.32 | CAC(H) | 0.4 | 0.15 | CGC(R) | 0 | 0 |
| CUA(L) | 1.4 | 0.21 | CCA(P) | 2.4 | 0.99 | CAA(Q) | 4.7 | 1.91 | CGA(R) | 2.7 | 2.59 |
| CUG(L) | 0 | 0 | CCG(P) | 0 | 0 | CAG(Q) | 0.2 | 0.09 | CGG(R) | 0.1 | 0.07 |
| AUU(I) | 33.8 | 1.94 | ACU(T) | 6.4 | 2.24 | AAU(N) | 18.7 | 1.92 | AGU(S) | 2.4 | 0.77 |
| AUC(I) | 1 | 0.06 | ACC(T) | 0.4 | 0.14 | AAC(N) | 0.8 | 0.08 | AGC(S) | 0 | 0 |
| AUA(M) | 20 | 1.86 | ACA(T) | 4.6 | 1.62 | AAA(K) | 6.8 | 1.81 | AGA(S) | 6.5 | 2.11 |
| AUG(M) | 1.5 | 0.14 | ACG(T) | 0 | 0 | AAG(K) | 0.7 | 0.19 | AGG(S) | 0 | 0 |
| GUU(V) | 5.8 | 1.99 | GCU(A) | 6.5 | 2.67 | GAU(D) | 4.7 | 1.94 | GGU(G) | 4.8 | 1.22 |
| GUC(V) | 0.1 | 0.03 | GCC(A) | 0.2 | 0.06 | GAC(D) | 0.2 | 0.06 | GGC(G) | 0.2 | 0.06 |
| GUA(V) | 5.1 | 1.75 | GCA(A) | 2.9 | 1.21 | GAA(E) | 5.4 | 1.87 | GGA(G) | 9.2 | 2.36 |
| GUG(V) | 0.7 | 0.24 | GCG(A) | 0.2 | 0.06 | GAG(E) | 0.4 | 0.13 | GGG(G) | 1.4 | 0.35 |

Average codons = 284

other similar codons. For example, the valine codons GTC and GTG were rare, whereas GTT and GTA's synonymous codons were prevalent (Table 2). In the mitogenome of *Leucoma salicis* [60] and *Eucrate crenata* [47] a similar tendency was observed. The pattern of codons for the frequently used amino acids like Phe, Leu, Ile, Asn and Ser of *C. acuta* are similar to most noctuoid mitogenomes [48, 61, 62].

### Transfer RNAs (*tRNA*) and ribosomal RNAs (*rRNA*)

The mitochondrial genome of *C. acuta* had 22 *tRNA* genes strewn over the circular genome. From the total, 14 *tRNAs* were present within H- strand, whereas the rest 8 were present within L- strand (Fig 1 and Table 1). The 22 *tRNAs'* aggregate sequence was 1,464 bp, 9.51% of the entire *mt* genome. Except for the *trnS1* (AGN) gene, which codes for the serine (Ser, S) amino acid, the secondary structures of the remaining *mt tRNA* of *C. acuta* were predicted; all genes revealed the characteristic clover leaf shape (Fig 2). The two *rRNA* genes *rrnL* of 1359 bp and *rrnS* of 781 bp are found on the L- strand, accounting for 13.90% of the *mt* genome (Fig 1). The length of 22 transfer RNA genes in the *C. acuta mt* genome ranged from 63 bp (trnR) to 71 bp (trnK), which is similar with the *mt* genomes of *L. suffusa* [51], *D. pyloalis* [54], and *C. menciana* [22]. Except for *trnS1* (AGN), which lacks the dihydrouridine (DHU) arm and forms a simple loop, all *tRNA* secondary structures resembled the conventional clover leaf shape (Fig 2). Many insect mitogenomes showed similar results, including *Bombyx mori* [63], *Actias selene* [64], *Spodoptera frugiperda* [65], *Spodoptera litura* [45], *Chrysochroa fulgidissima* [66], *B. zonata* [67] *Idioscopus nitidulus* [68] *Acronicta rumicis* [69] and *Eudocima salamina* [48].

There was an unusual (G-U and U-U) mismatch pair in the tRNA genes. Ten of the *tRNA* genes were found to contain 14 G-U mismatches in their secondary structures, forming a weak bondThe amino acid acceptor stems of *tRNALeu*(UUR), *tRNAAla*, *tRNASer1*, and *tRNASer2* contained five U-U mismatches (Fig 2). Sun *et al.* [60] and Riyaz *et al.* [48] also observed a similar trend in *L. salicis and E. salamina* respectively. The control (AT-rich) region varies in

**Fig 2. Predicated secondary clover-leaf structures for the 22 *tRNA* genes of *Chrysodeixis acuta*.** The tRNAs are labled with abbreviation of their corresponding amino acids below each tRNA gene structure. Arms of tRNAs (clockwise from top) are the amino acid acceptor arm, TYC arm, the anticodon arm, and dihydrouridine (DHU) arm.

size across members of the Noctuidae family, although it includes comparable sequence components in most species. In many Lepidopteran species, the ATAGA motif found at the start of the control region is a conserved motif sequence. Similarly, Li *et al.* [51], Dai *et al.* [22], Chen *et al.* [69] and Liu *et al.* [70] have also reported the ATAGA motif in the control region followed by the Poly-T stretch and microsatellite A/T repeats elements along with Poly-A stretch at the end of the control region.

## Overlapping and intergenic spacer regions

The whole circular mitochondrial genome map of *C. acuta* has six overlapping sections ranging from 1 to 20 bp with a total length of 43 bp. The overlapping area between the *trnL* and *rrnL* genes is the longest one measuring 20 bp. Other overlapping regions include 7 bp between *trnC* and *trnW*, *cox1* and *trnY*; 6 bp between *atp6* and *atp8*, 2 bp between *trnQ* and *trnI*, and a single base pair between *trnA* and *nad3* genes (Table 1). The *mt* genome of *C. acuta* comprises 18 intergenic spacer sequences, ranged 1 to 111 bp which constituted a total 427 bp. The longest intergenic spacer was 111 bp, present between *nad2* and *trnW* is an A/T rich

```
Diachrysia nadeja        TGTACTAAAAATAAATCAAT
Chrysodeixis acuta         ATACTAAAAATAAATYAATTTATAATAAAAA
Trichoplusia ni            ATACTAAAAATAAATTAAT
Ctenoplusia limbirena    TTATACTAAAAATAAATTAAT
Ctenoplusia albostriata  TATACTAAAAATAAATTAAT
```

**Fig 3. Alignment of the intergenic spacer region between *trnS2* (UCN) and *nad1* of several lepidopteran insects of Plusiinae subfamily.** ATACTAA motif is underlined and the TACTAAAAATAAAT is shaded.

region. The largest intergenic spacer sequences are 74 bp and 72 bp situated between *trnQ* and *nad2* genes, and *trnH* and *nad4* genes respectively. The other intergenic spacers are less than 50 bp size (Table 1). The spacer region between *trnS2* (UCN) and *nad1* includes two motifs 'ATACTAA' and 'TACTAAAAATAAAT' of 7 and 14 bp respectively (Fig 3). The intergenic spacer region between the genes *trnS2* (UCN) and *nad1* includes the 'ATACTAA' motif, which is commonly present in most lepidopteran species; even though the size of the intergenic spacer region varies [52, 69–71]. This spacer region is generally considered a constitutive synapo-morphic feature of the lepidopteran mitochondrial genomes because this region won't present in non-lepidopteran insects species [72]. The similar motif were located in the *Dysgonia stuposa* [73], *L. salicis* [60], *C. menciana* [22] and *Ctenoptilum vasava* [52]. The motif 'TACTAAAAATAAAT' was located in the intergenic spacer region between the genes *trnS2* (UCN) and *nad1* of the *C. acuta*, were also common in lepidopteran species (Fig 3). However, this motif was not reported in other than lepidopteran families.

## The control region

The control region (AT-rich) of *C. acuta mt* genome comprises 339 bp is positioned between *rrnS* and *trnM* genes. It is one of the longest mitochondrial non-coding region which cover about 2.20% of the whole mitochondrial genome and contains 93.51% AT nucleotides. There have been no reports of noticeable extensive repeats in the AT-rich control region. However, many short repetitive sequences are distributed across the region, including the ATAGA motif, followed by a 19-bp Poly-T stretch, a microsatellite-like $(AT)_{10}$, and a 9-bp poly-A stretch downstream of *trnM*. (Fig 4). Similar sequence elements were found in the mitochondrial genomes of lepidopteran species, *Laelia suffusa* [51], *Cerura menciana* [22], *Hestina persimilis* and *Hestinalis nama* [74].

## Phylogenetic analysis

The phylogenetic tree of thenoctuidae family comprising 55 Lepidoptera species based on 13 PCGs nucleotide sequence datasets constructed by utilizing Maximum Likelihood (ML), and Bayesian Inference (BI) methods. The phylogenetic tree, as shown in Fig 5, revealed that the *mt* genome of *C. acuta* taxonomic status was closest with *T. ni* (GenBank accession No. MK 714850), *C. limbirena* (GenBank accession No. KM 244665), and *C. albostriata* (GenBank

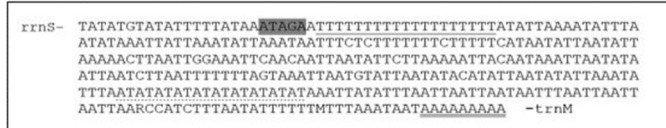

**Fig 4. Features presents A+T rich region of *Chrysodeixis acuta*.** The ATAGA motif is shaded, Poly-T strand is underlined, Poly-A stretch is double underlined; the single microsatellite A/T repeat sequence is dotted underlined.

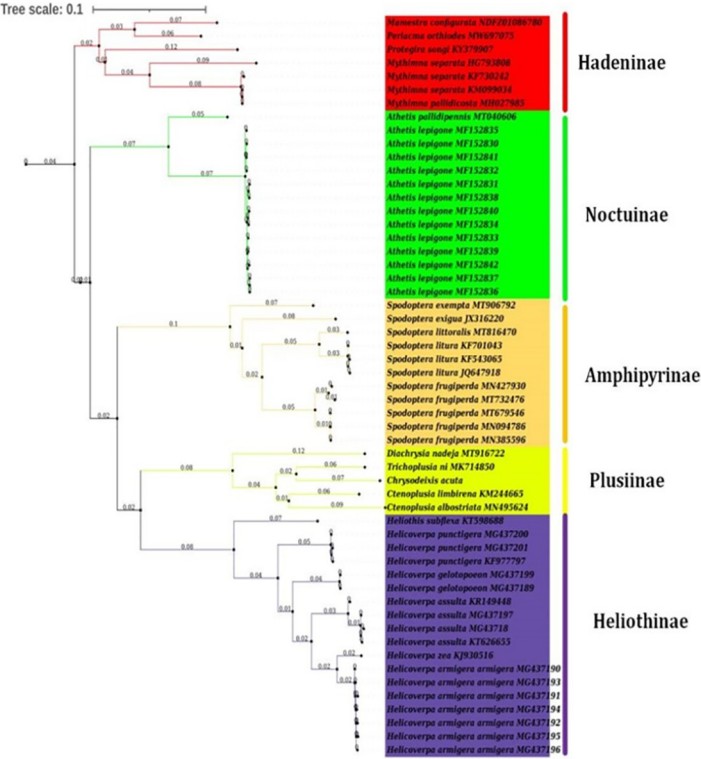

**Fig 5. Phylogenetic tree of 55 species from Noctuidae family species including *Chrysodeixis acuta*.** The analysis obtained from Bayseain Inference and Maximum Likelihood based on concatenated data of 13 PCGs genes. The numbers at nodes indicate ML bootstrap values probabilities. Accession numbers are given for species obtained from GenBank.

accession No. MN 495624) from the same sub-family *Plusiinae*. Hadeninae, Noctuinae, Amphipyrinae, Plusiinae, and Heliothinae sub-families of the Noctuidae family are closely related. The sub-family of *C. acuta*, *Plusiinae* is closely related to the *Heliothinae and Amphipyrinae*. However, the *Hadeninae* and *Noctuinae* are sister groups from the same superfamily (Fig 5). According to the overall findings of the phylogenetic analysis, species that belong to the same family but different subfamilies are placed together.

The phylogenetic analysis discovered that *C. acuta* is most closely correlated to the cabbage looper, *Trichoplusia ni*, from the same subfamily Plusiinae, notably in mt genome similarities, 13 PCGs, 22 tRNAs, and two rRNAs. However, the size of the control region appears to be different [43]. The five sub-families comprising *Hadeninae*, *Noctuinae*, *Amphipyrinae*, *Plusiinae*, and *Heliothinae* from the *Noctuidae* family are closely related. The sub-family of *C. acuta*, *Plusiinae* is closely related to the *Heliothinae and Amphipyrinae*. However, the *Hadeninae* and *Noctuinae* are sister groups from the same superfamily (Fig 5). The phylogenetic tree confirmed that the previously characterized species *C. albostriata* [49], *D. nadeja* [50], and *T. ni* [43] belonged to the sub-family Plusiinae and are related to the *C. acuta* which confirms that the *C. acuta* also belonged to the Plusiinae sub-family. The phylogenetic relationships were reconstructed based on the concentrated data of the 13 PCGs, which supports the traditional morphology-based view of the relationship within the Noctuidae family. The phylogenetic relationships were reconstructed based on the concentrated data of the 13 PCGs, which supports the traditional morphology-based view of the relationship within the Noctuidae family.

## Nucleotide composition

The mitochondrial genomes of the insects generally reflect real strand discrimination in nucleotide compositions [75]. The strand dissimilarity is measured as the AT and GC-skews [30]. The J-strand displayed strand bias with negative GC-skew and positive AT-skew pattern (S1 Fig) in a comparative study of nucleotide percentages *vs* skewness of the 55 Noctuidae family members, including *C. acuta*. The relative synonymous codon usages (RUSC) were evaluated to identify the preference for a specific synonymous (Table 2). The codon usages pattern of *C. acuta* suggests that the two-fold and four-fold degenerate codons overuse the A/T at the third codon position (S2 Fig). Two-fold and four-fold degenerate codon usage was obviously A/T biased over G/C in the third position [76].

## Gene evolution rate

For the analysis of gene evolution rate, the rate of non-synonymous substitution (Ka, pi modified), synonymous substitution (Ks, pi modified), and the Ka/Ks ratio, as well as the Jukes-Cantor adjusted Ka/Ks (JKa/JKs) ratio, of 13 PCGs from 55 mitogenome sequences from Noctuidae family species, including *C. acuta* was used. Based on synonymous and non-synonymous nucleotide substitution analysis, the Ka/Ks values of the 13 PCGs of 55 Noctuidae species ranged from 0.0463 to 1.0947 (Table 3). The Ka/Ks values of only two genes *nad2* and *cytB* were more than one, 1.0947 and 1.0654, respectively, indicating the highest evolutionary rates. While *cox1* recorded lowest (0.0463) among the 13 PCGs. Similarly, Jukes-Cantor adjusted Ka/Ks also showed that the *cytB* and *nad2* had highest evolutionary rate with JKa/JKs values 2.0443 and 1.6206, respectively. However, *cox1* and *atp6* recorded the lowest JKa/JKs values of 0.0349 and 0.0916, respectively. Overall ratios of Ka/Ks and JKa/JKs for the genes *cox1*, *cox2*, *atp6*, *nad3*, *nad4*, and *nad4L* were less than 0.5, indicating that these genes in the Noctuidae family evolved under purifying selects (S3 Fig). Though this gene has a relatively slow evolutionary rate, this could be used as the candidate barcoding marker for species identification in Noctuidae family. Nucleotide substitution in the mitogenome indicates the evolution at the molecular level [77]. Nucleotide substitution was previously thought to represent a directional bias between various genes in the mitochondrial genome of insects, according to previous research by Cameron [39].

**Table 3. Rates of non-synonymous substitutions (Ka, pi modified), synonymous substitutions (Ks, pi modified) and the Ka/Ks ratio as well as Jukes-Cantor adjusted Ka/Ks (JKa/JKs) ratio in each PCG of 55 mitogenome sequences of Noctuidae family species including *Chrysodeixis acuta* from this study.**

| Protein-coding genes | Rates of non-synonymous substitutions (Ka) | Rates of synonymous substitutions (Ks) | Ka/Ks ratio | Rates of non-synonymous substitutions Jukes-Cantor adjusted J(Ka) | Rates of synonymous substitutions Jukes-Cantor adjusted J(Ks) | JKa / JKs ratio |
|---|---|---|---|---|---|---|
| *nad2* | 0.33444 | 0.30550 | 1.09473 | 0.65868 | 0.40644 | 1.62061 |
| *cox1* | 0.01457 | 0.31435 | 0.04635 | 0.01479 | 0.42349 | 0.03492 |
| *cox2* | 0.05820 | 0.33466 | 0.17391 | 0.11659 | 0.46371 | 0.25143 |
| *atp8* | 0.15659 | 0.21912 | 0.71463 | 0.18637 | 0.26938 | 0.69185 |
| *atp6* | 0.0402 | 0.3316 | 0.121261 | 0.04187 | 0.4573 | 0.0916 |
| *cox3* | 0.27558 | 0.37520 | 0.73449 | 0.76770 | 0.55221 | 1.39023 |
| *nad3* | 0.10409 | 0.38134 | 0.27296 | 0.18199 | 0.56431 | 0.32250 |
| *nad6* | 0.27856 | 0.28836 | 0.96601 | 0.43713 | 0.37632 | 1.16159 |
| *nad5* | 0.1939 | 0.2619 | 0.740482 | 0.45273 | 0.3326 | 1.3611 |
| *nad4* | 0.04544 | 0.24551 | 0.18508 | 0.04748 | 0.30501 | 0.15567 |
| *nad4L* | 0.10840 | 0.24782 | 0.43741 | 0.17589 | 0.31478 | 0.55877 |
| *cytB* | 0.40883 | 0.38370 | 1.06549 | 1.15181 | 0.56342 | 2.04432 |
| *nad1* | 0.19112 | 0.26457 | 0.72238 | 0.49746 | 0.33763 | 1.47339 |

## Conclusions

Many agricultural pests and economically important insects are found in the order Lepidoptera. Most economically important defoliator pests belong to the Noctuidae family. The mitogenome was utilised to elucidate the evolutionary position of Lepidoptera at several taxon levels, especially for Noctuoidea [78]. The mitogenomes of the Noctuidae family have been studied extensively; however, very few reports are available on the *Plusiinae* subfamily. The first complete *mt* genome sequence of *Chyrsodeixis acuta* is revealed in this work. The circular mitogenome of *C. acuta* is 15,399 bp length and contains 37 sequence elements. It comprises 13 protein-coding genes, *ATPase* genes, *Cytochrome B* gene, *cytochrome c oxidase* gene, *NADH dehydrogenase* gene, two ribosomal RNA genes, 22 transfer RNA genes, and a control region. The complete *mt* genome adds to the genetic richness of this species and elucidates crucial information for further evolutionary and phylogenetic studies of the Noctuidae family. In addition, other potential pests of the genus *Chrysodeixis*, such as *Chrysodeixis chalcites*, *Chrysodeixis eriosoma*, and *Chrysodeixis includens*, infesting various crops, would benefit from the mitochondrial genomic organization of *Chyrsodeixis acuta*.

## Supporting information

**S1 Fig. AT% vs AT-skew and GC% vs GC-skew in the 63 Noctuidae family species including *Chrysodeixis acuta* from this study.** Values are calculated on J-strands for full length of mt genomes. The X-axis provides the skews values, while the Y axis provides the A+T/G+C values. Names of species are colored according to their taxonomic placement.
(TIF)

**S2 Fig. The AT content percentage of 0-fold degenerate sites, 2-fold degenerate sites and 4-fold degenerate sites in each protein coding gene of 55 mitochondrial genome sequences of Noctuidae family species including *C. acuta* from this study.** The black line with short line on the top of each bar represents the standard deviation value (SD).
(TIF)

**S3 Fig. Ratio of non-synonymous substitutions (Ka, pi modified) & synonymous substitutions (Ks, pi modified) (Ka/Ks) as well as Jukes-Cantor adjusted (JKa/JKs) ratio in each PCG of 55 mitogenome sequences of Noctuidae family species including *C. acuta* from this study.**
(TIF)

## Acknowledgments

The authors are thankful to the Head of the Division of Entomology, ICAR-Indian Agricultural Research Institute, New Delhi, India, and Dr. P. R. Shashank, Division of Entomology, ICAR-IARI, New Delhi, India, for insect species confirmation. We also gratefully acknowledge the editors and anonymous reviewers for their valuable suggestions and comments.

**Permissions**: The researchers were granted permission to collect the insect sample as part of the routine monitoring of the experimental trials for emerging and invasive pest species by the Director ICAR-Directorate of Onion and Garlic Research, Pune, Maharashtra, India.

## Author Contributions

**Conceptualization:** Soumia P. S., Guru Pirasanna Pandi G., Anandhan S.

**Data curation:** Soumia P. S., Dhananjay V. Shirsat, Ram Krishna, Guru Pirasanna Pandi G., Jaipal S. Choudhary, Naiyar Naaz, Anandhan S.

**Formal analysis:** Soumia P. S., Dhananjay V. Shirsat, Ram Krishna, Guru Pirasanna Pandi G., Jaipal S. Choudhary, Naiyar Naaz, Anandhan S.

**Funding acquisition:** Major Singh.

**Methodology:** Soumia P. S., Dhananjay V. Shirsat, Guru Pirasanna Pandi G., Anandhan S.

**Project administration:** Soumia P. S., Anandhan S., Major Singh.

**Resources:** Major Singh.

**Software:** Soumia P. S.

**Supervision:** Soumia P. S., Anandhan S., Major Singh.

**Validation:** Soumia P. S.

**Visualization:** Soumia P. S.

**Writing – original draft:** Soumia P. S., Dhananjay V. Shirsat, Ram Krishna, Guru Pirasanna Pandi G., Jaipal S. Choudhary, Naiyar Naaz, Karuppaiah V., Pranjali A. Gedam, Anandhan S.

**Writing – review & editing:** Soumia P. S., Dhananjay V. Shirsat, Ram Krishna, Guru Pirasanna Pandi G., Karuppaiah V., Pranjali A. Gedam, Anandhan S.

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
