## [Decision Letter · Decision Letter 0]

20 Jul 2022

PONE-D-22-12683Unfolding the mitochondrial genome structure of green semilooper (Chrysodeixis acuta Walker): an emerging pest of onion (Allium cepa L.)PLOS ONE

Dear Dr. P S,

Thank you for submitting your manuscript to PLOS ONE. After careful consideration, we feel that it has merit but does not fully meet PLOS ONE’s publication criteria as it currently stands. Therefore, we invite you to submit a revised version of the manuscript that addresses the points raised during the review process.

We look forward to receiving your revised manuscript.

Kind regards,

Muhammad Faisal Shahzad, Ph.D.

Academic Editor

PLOS ONE

Journal Requirements:

“The present study is a part of in-house project, Reference code “IXX14062." The financial support is given by ICAR- Directorate of Onion and Garlic Research, Pune”

“The authors are thankful to the Director ICAR-Directorate of Onion and Garlic Research, Pune, Maharashtra, India, for providing funds. The present study is a part of in-house project, Reference code “IXX14062."”

“The present study is a part of in-house project, Reference code “IXX14062." The financial support is given by ICAR- Directorate of Onion and Garlic Research, Pune”

5. Please upload a new copy of Figure 5 as the detail is not clear. Please follow the link for more information: https://blogs.plos.org/plos/2019/06/looking-good-tips-for-creating-your-plos-figures-graphics/" https://blogs.plos.org/plos/2019/06/looking-good-tips-for-creating-your-plos-figures-graphics/

Additional Editor Comments:

I agree with the reviewers that the MS should be revised as per suggestions recommended, The authors should carry out careful revision. If this is not done carefully, it is likely to lead to rejection after review.

Reviewers' comments:

Reviewer's Responses to Questions

**Comments to the Author**

1. Is the manuscript technically sound, and do the data support the conclusions?

Reviewer #1: Yes

Reviewer #2: Yes

2. Has the statistical analysis been performed appropriately and rigorously? 

Reviewer #1: N/A

Reviewer #2: Yes

3. Have the authors made all data underlying the findings in their manuscript fully available?

Reviewer #1: Yes

Reviewer #2: Yes

4. Is the manuscript presented in an intelligible fashion and written in standard English?

Reviewer #1: Yes

Reviewer #2: Yes

5. Review Comments to the Author

Reviewer #1: In this MS，the mitochondrial genome of a onion pest Chrysodeixis acuta was identified and the phylogenetic study was performed. The results will provide some useful information for the research of insect mitochondrial genomes.

1.How many samples were selected？the stage of the larval? And the mitochondrial genome was sequenced only once?

2.The quality of the figures Is not goog and need improvement.

3.The scientific name of the species should be italized and the format of references Is insonsistent.

4. The English should be further improved.

Reviewer #2: I would only suggest a widening of literature references, also considering some more recent works that came out, and hence a deeper discussion.

The manuscript can be accepted for publication after revising for minor typographical and grammatical revisions.

6. PLOS authors have the option to publish the peer review history of their article (what does this mean?). If published, this will include your full peer review and any attached files.

Reviewer #1: No

Reviewer #2: No

---

## [Author Response · Author response to Decision Letter 0]

4 Aug 2022

Reviewer comment

Reviewer 1

Comment 1: How many samples were selected? The stage of the larval? And the mitochondrial genome was sequenced only once?

Author Response: Laboratory reared fully grown fourth instar larvae of Chrysodeixis acuta was used for the mitogenome analysis. Single sample was sequenced. As per the reviewer’s suggestion, necessary changes has been made in the materials and methods section of the revised manuscript.

Comment 2: The quality of the figures is not good and need improvement.

Author Response: Quality of the figures are improved and included in the revised manuscript.

Comment 3: The scientific name of the species should be italicized and the format of references is inconsistent 

Author Response: Scientific name of the species were italicized and the formatting of the references have been done as per the journal norms.

Comment 4: The English should be further improved 

Author Response: As per the reviewer’s suggestion, necessary changes has been made in the revised manuscript.

Reviewer 2

Comment 1: I would only suggest a widening of literature references, also considering some more recent works that came out, and hence a deeper discussion.

Author Response: As per the reviewer’s suggestion, discussion section has been elaborated with recent publications in the revised manuscript. 

Comment 2: Revise for minor typographical and grammatical errors.

Author Response: Both minor typographical and grammatical errors has been rectified in the revised manuscript.

---

## [Editor Report · Decision Letter 1]

15 Aug 2022

Unfolding the mitochondrial genome structure of green semilooper (Chrysodeixis acuta Walker): an emerging pest of onion (Allium cepa L.)

PONE-D-22-12683R1

Dear Dr. SOUMIA P S, 

We’re pleased to inform you that your manuscript has been judged scientifically suitable for publication and will be formally accepted for publication once it meets all outstanding technical requirements.

Kind regards,

Muhammad Faisal Shahzad, Ph.D.

Academic Editor

PLOS ONE
---

## [Editor Report · Acceptance letter]

19 Aug 2022

PONE-D-22-12683R1 

Unfolding the mitochondrial genome structure of green semilooper (*Chrysodeixis acuta* Walker): an emerging pest of onion (*Allium cepa * L.) 

Dear Dr. P.S.:

I'm pleased to inform you that your manuscript has been deemed suitable for publication in PLOS ONE. Congratulations! Your manuscript is now with our production department. 

Kind regards, 

on behalf of

Dr. Muhammad Faisal Shahzad 

Academic Editor

PLOS ONE